# Phylogenetic and functional diversity among *Drosophila*-associated metagenome-assembled genomes

Aaron A. Comeault,[1] Alberto H. Orta,[1] David B. Fidler,[1] Tobias Nunn,[1] Amy R. Ellison,[1] Tayte A. Anspach,[2] Daniel R. Matute[2]

**ABSTRACT** Host-associated microbial communities can mediate interactions between their hosts and biotic and abiotic environments. While much work has been done to document how microbiomes vary across species and environments, much less is known about the functional consequences of this variation. Here, we test for functional variation among drosophilid-associated bacteria by conducting Oxford Nanopore long-read sequencing and generating metagenome-assembled genomes (MAGs) from communities associated with six species of drosophilid flies collected from "anthropogenic" environments in North America, Europe, and Africa. Using phylogenetic analyses, we find that drosophilid flies harbor a diverse microbiome that includes core members closely related to the genera *Gilliamella*, *Orbus*, *Entomomonas*, *Dysgonomonas*, and others. Comparisons with publicly available bacterial genomes show that many of these genera are associated with phylogenetically diverse insect gut microbiomes. Using functional annotations and predicted secondary metabolite biosynthetic gene clusters, we show that MAGs belonging to different bacterial orders and genera vary in gene content and predicted functions, including metabolic capacity and how they respond to environmental stressors. Our results provide evidence that wild drosophilid flies harbor phylogenetically and functionally diverse microbial communities. These findings highlight a need to quantify the abundance and function of insect-associated bacteria from the genera *Gilliamella*, *Orbus*, *Entomomonas*, and others on the performance of their insect hosts across diverse environments.

**IMPORTANCE** While much attention has been given to catalogue the taxonomic diversity intrinsic to host-associated microbiomes, much less is known about the functional consequences of this variation, especially in wild, non-model host species. In this study, we use long-read sequencing to generate and analyze 103 high-quality metagenome-assembled genomes from host-associated bacterial communities from six species of wild fruit fly (*Drosophila*). We find that the genomes of drosophilid-associated bacteria possess diverse metabolic pathways and biosynthetic gene clusters that are predicted to generate metabolites involved in nutrition and disease resistance, among other functions. Using functional gene predictions, we show that different bacterial lineages that comprise the insect microbiome differ in predicted functional capacities. Our findings highlight the functional variation intrinsic to microbial communities of wild insects and provide a step towards disentangling the ecological and evolutionary processes driving host–microbe symbioses.

**KEYWORDS** metagenomics, symbiosis, *Drosophila*, insects, microbiome, MAGs, Orbaceae

**Peer Reviewer** Irene L. G. Newton, Indiana University, Bloomington, Bloomington, Indiana, USA

Address correspondence to Aaron A. Comeault, a.comeault@bangor.ac.uk.

The authors declare no conflict of interest.

See the funding table on p. 18.

Host-associated microbial communities (microbiomes) affect their hosts' biology and how those hosts interact with the environment in diverse ways. Despite the clear effects that microorganisms can have on the performance and fitness of their

hosts, many studies of microbiomes associated with wild non-model organisms have characterized variation in the microbiome using 16S rRNA gene community profiling (1). While 16S rRNA gene community profiling has provided important insight into how biotic and abiotic variation can affect microbiome composition (2), we know less about how specific microorganisms—across diverse hosts and environments—affect functional or performance traits of the microbiome and host species. Estimating functional traits of specific microorganisms is one way that we can begin to generate a mechanistic understanding of how members of microbial communities affect host performance.

Insects—the most diverse group of animals—provide valuable ecosystem services (e.g., pollination and nutrient cycling) (3), are important agricultural pests, and vector a range of diseases, some with economic impacts of billions of US dollars per year (4). Insect-associated microorganisms can affect these processes by detoxifying environmental and dietary toxins (5), moderating vectorial capacity (6), conferring resistance to infection of their hosts (7, 8), and even contributing to successful biological invasions (9). Among insects, fruit flies from the genus *Drosophila* (particularly *Drosophila melanogaster*) have emerged as a model system used to study host–microbe interactions (10–12). Research on host–microbiome interactions in *D. melanogaster* has largely focused on the effects of interactions with bacteria from the genera *Acetobacter* and *Lactobacillus*. For example, *Acetobacter* have been shown to promote larval development when larvae are raised on deficient diets, suggesting links between bacteria and host nutrition (13); mate choice in adult *D. melanogaster* can be influenced by *Lactobacillus* acquired through diet (14); and experimental manipulation of *Acetobacter* and *Lactobacillus* together can lead to rapid evolution in *D. melanogaster*, indicating that these bacteria are a source of selection that can contribute to local adaptation (15, 16). However, the microbial communities of wild versus lab-reared *Drosophila* differ significantly, and *Acetobacter* and *Lactobacillus* can be rare in wild *Drosophila* (10, 17). Abundances of these genera have also been shown to vary across laboratory strains of 11 *Drosophila* species (18). The few studies that estimate microbial diversity in wild *Drosophila* have shown that they harbor a more diverse microbiome than laboratory populations (10, 19). Common members of wild *Drosophila* microbiomes include bacteria from the orders Bacteriodales and Pseudomonadales and families Orbaceae, Enterobacteriaceae, and Enterococcaceae (10, 19, 20). Therefore, while laboratory experiments in *D. melanogaster* provide evidence that symbiotic or commensal bacteria can affect diverse aspects of their host's biology, differences between the bacteria that are the focus of laboratory studies and those reported in wild *Drosophila* highlight a need to better understand assembly rules and functional roles of members of wild *Drosophila* microbiomes.

Despite significant variation among the microbiomes of *Drosophila* species and individuals, the composition of wild *Drosophila* microbiomes suggests that these communities are not randomly assembled from a common pool of microorganisms, and patterns of co-occurrence suggest negative interactions between certain bacterial taxa (19). Recent metagenomic analyses also provide correlative evidence for functional variation among the gut bacteria associated with three mushroom-feeding *Drosophila* species (20). However, we still have a poor understanding of the functional traits possessed by specific *Drosophila*-associated bacteria. Quantifying functional variation among *Drosophila*-associated bacteria would facilitate clearer hypotheses and predictions of the impacts of host-associated bacteria on host performance and fitness. It would also allow researchers to leverage the ecological and evolutionary diversity found among species of *Drosophila* (21–24) to better understand the processes and mechanisms that shape host–microbiome interactions in the wild.

The gut microbiomes of honeybees (genus *Apis*) are arguably the best functionally characterized insect microbiome (25, 26). For example, functional work on *Gilliamella* and *Snodgrassella*—core members of the bee microbiome—has shown how these bacteria possess complementary metabolic pathways and genes that affect gut colonization

(27). *Gilliamella apicola* is also capable of metabolizing toxic sugars (28), and screens for biosynthetic gene clusters possessed by *G. apis* have identified unique ribosomally synthesized, post-translationally modified peptides (RiPPs) that protect *A. mellifera* from infection by the pathogen *Melissococcus plutonius* (7). These examples illustrate how identifying core microbiome taxa, along with using "bottom-up" genome annotation approaches to predict their functional capacities, can be important first steps towards identifying genes that mediate microbe–microbe interactions and affect host fitness.

In this study, we mine whole-organism long-read Oxford Nanopore Technologies (ONT) sequences to study phylogenetic and functional diversity among bacteria associated with six species of drosophilid flies collected from sample locations in the UK, the USA, and São Tomé. For each host sample, we identified and classified bacterial sequences from ONT sequences generated from individual wild-caught flies to estimate microbiome diversity and core microbial taxa. We then generated metagenome-assembled genomes (MAGs) for each sample and classified MAGs against the Genome Taxonomy Database. Finally, we annotated high-quality MAGs and compared functional predictions among taxonomic groups. We found that wild *Drosophila* are host to a diverse microbiome, including putatively "core" taxa that are shared among species and sample locations. These core taxa are related to bacteria from the orders Enterobacterales, Pseudomonodales, and Bacteriodales that have been sequenced from other insects, including *Gilliamella* and *Frischella* bacteria that are core members of the *Apis* (honeybees) and *Bombus* (bumble bees) gut microbiome. We also find that *Drosophila*-associated MAGs from different genera vary in gene content, predicted functional enrichments, and predicted ability to produce secondary metabolic products.

## RESULTS

### Bacterial diversity across sampled reads

By mining whole-organism sequence reads with Kraken 2, we recovered a median of 198,036 bacterial reads (range: 30,630–5,574,666) spanning a median of 765.21 Mb of sequence (range 197.91–8,829.46 Mb) per individual drosophilid fly (see Table S1 and Fig. S1 at https://doi.org/10.5281/zenodo.14173039). Across all samples, sequence reads were uniquely assigned to 12 phyla, 23 classes, 62 orders, 114 families, and 254 genera of bacteria. Gammaproteobacteria, Alphaproteobacteria, Bacilli, and Mollicutes were each the most abundant class of bacteria in sequences obtained from 22, four, three, and two individuals, respectively; and Gammaproteobacteria, Alphaproteobacteria, and Bacilli were the second most abundant classes of bacteria in individuals where they were not the most abundant, except for one individual in which Epsilonproteobacteria was the second most abundant class (13.4% of reads) and three individuals in which Flavobacteria was the second most abundant class (13.1% to 26.0% of reads) (Fig. 1A; also see Fig. S2 at https://doi.org/10.5281/zenodo.14173039). The percentage of reads assigned to the most abundant class of bacteria per individual ranged from 25.0% to 90.2%, while the percentage of reads assigned to the second most abundant class of bacteria per individual ranged from 3.9% to 43.1% (Fig. 1A). Six genera of bacteria were represented by at least 1% of sequences classified as bacteria in 16 or more of the individual drosophilids we sampled. These genera were *Vibrio*, *Pseudomonas*, *Acinetobacter*, *Clostridium*, *Bacillus*, and *Halomonas*. Reads assigned to *Acetobacter* and *Lactobacillus*—the two genera of bacteria that are the focus of laboratory research in *Drosophila melanogaster*—were relatively rare across our samples, with median proportions of reads per sample of 0.18% (90% empirical quantile: 0.03% to 8.25%) and 0.22% (0.04% to 2.51%), respectively. Inspecting rarefaction curves showed that taxonomic classification across reads was saturated for >50% of sampled *Drosophila*; however, taxon sampling was not saturated in individuals where we recovered fewer than ~100,000 bacterial reads (see Fig. S3 and Table S1 at https://doi.org/10.5281/zenodo.14173039).

We summarized the differences in microbial taxa abundances among samples by first conducting a multidimensional scaling analysis. The first principal coordinate axis (PCoA) resulting from this analysis explained 29% of variation in leading log2-fold changes in

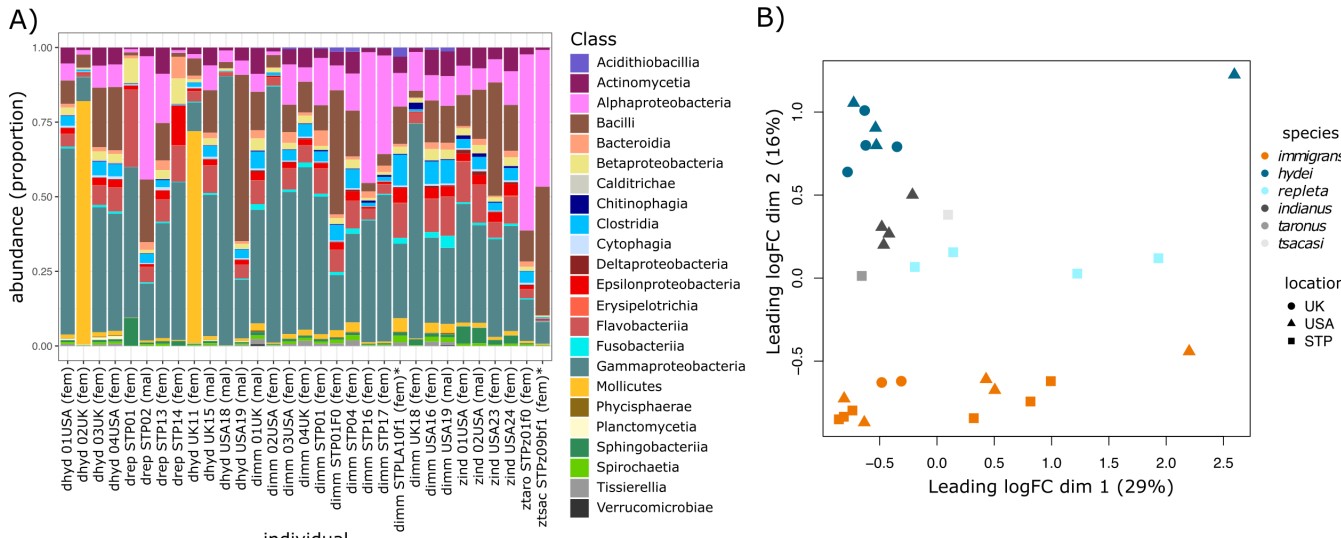

**FIG 1** Bacterial sequence reads derived from whole-organism extraction and Nanopore MinION sequencing represent diverse taxa belonging to 23 classes (A). Differences in the relative abundances of microbial taxa among samples were affected by the number of sequences classified as bacterial (B; dim 1) and by species-level differences in the microbial community (B; dim 2). In panel A, individual names include details of the species (dhyd = *D. hydei*; drep = *D. repleta*; dimm = *D. immigrans*, zind = *Z. indianus*, ztaro = *Z. taronus*, and ztsac = *Z. tsacasii*), location (USA, UK, or STP [São Tomé and Principe]), and sex (fem = female, mal = male). Two individuals in the dataset were F1 offspring from wild-caught females and are indicated with an asterisk (*) in panel A.

bacterial read abundances among samples (Fig. 1B); however, PCoA dimension 1 scores are correlated with the number of bacterial sequences recovered from a sample (Pearson's rho = 0.63; $P < 0.00015$). The second PCoA accounted for 16% of variation among samples and differentiated them based on host species rather than the location the hosts were collected from (Fig. 1B). Redundancy analysis on the taxonomic matrix revealed that the interaction between the number of bacterial sequence reads and the host species significantly affected bacterial diversity and abundance (model $R^2 = 81.35\%$; permutation test: $F_{9,21} = 15.54$; $P = 0.003$; see Fig. S4 at https://doi.org/10.5281/zenodo.14173039). Classification and analysis of bacterial sequences therefore allowed us to quickly identify diverse bacterial communities that varied among host species. However, we do not explore community diversity further because the ONT sequencing we conducted is PCR-free whole-genome shotgun sequencing. Therefore, variation in read abundance could be affected by factors, such as differences in taxonomic abundance, genome size, and sequence classification accuracy among bacterial taxa and across their genomes.

## Bacterial diversity across MAGs

Across all samples, we assembled 143,751 contigs that were then binned into 366 MAGs (see Table S1 at https://doi.org/10.5281/zenodo.14173039). We retained 103 "focal" MAGs after filtering for CheckM completeness scores greater than 45% and contamination less than 10%. We were unable to recover MAGs from six of our 31 sampled host individuals, and both the number of contigs and MAGs assembled from a sample were correlated with the amount of bacterial sequence data recovered from the whole-organism sequences (Kendall's tau = 0.65 and 0.58, respectively; both $P < 1 \times 10^{-5}$; see Fig. S5 at https://doi.org/10.5281/zenodo.14173039).

Largely consistent with read-level classifications reported by Kraken 2, the most abundant classes of bacteria across the 103 focal MAGs were Gammaproteobacteria (55 MAGs), Bacteroidia (21 MAGs), Bacilli (11 MAGs), Alphaproteobacteria (10 MAGs), Clostridia (four), and Campylobacteria (two). Using GTDB-Tk taxonomies, we identified seven orders that were represented by at least five MAGs: Enterobacterales (N = 27), Pseudomonodales (N = 21), Bacteriodales (N = 14), Lactobacillales (N = 9), Acetobaterales

(*N* = 7), Flavobacteriales (*N* = 5), and Burkholderiales (*N* = 5) (Fig. 2A). Three of these orders—Enterobacterales, Pseudomonodales, and Bacteriodales—were represented by multiple MAGs assembled from at least two host species, and within at least one of those hosts, they were assembled from samples collected at multiple locations (Fig. 2B through D). Within these three orders, seven genera were represented by MAGs assembled from *D. hydei*, *D. repleta*, and *D. immigrans* (our most widely sampled species) and/or from multiple geographic locations (Fig. 3). Two MAGs were classified as *Acetobacter* spp.,

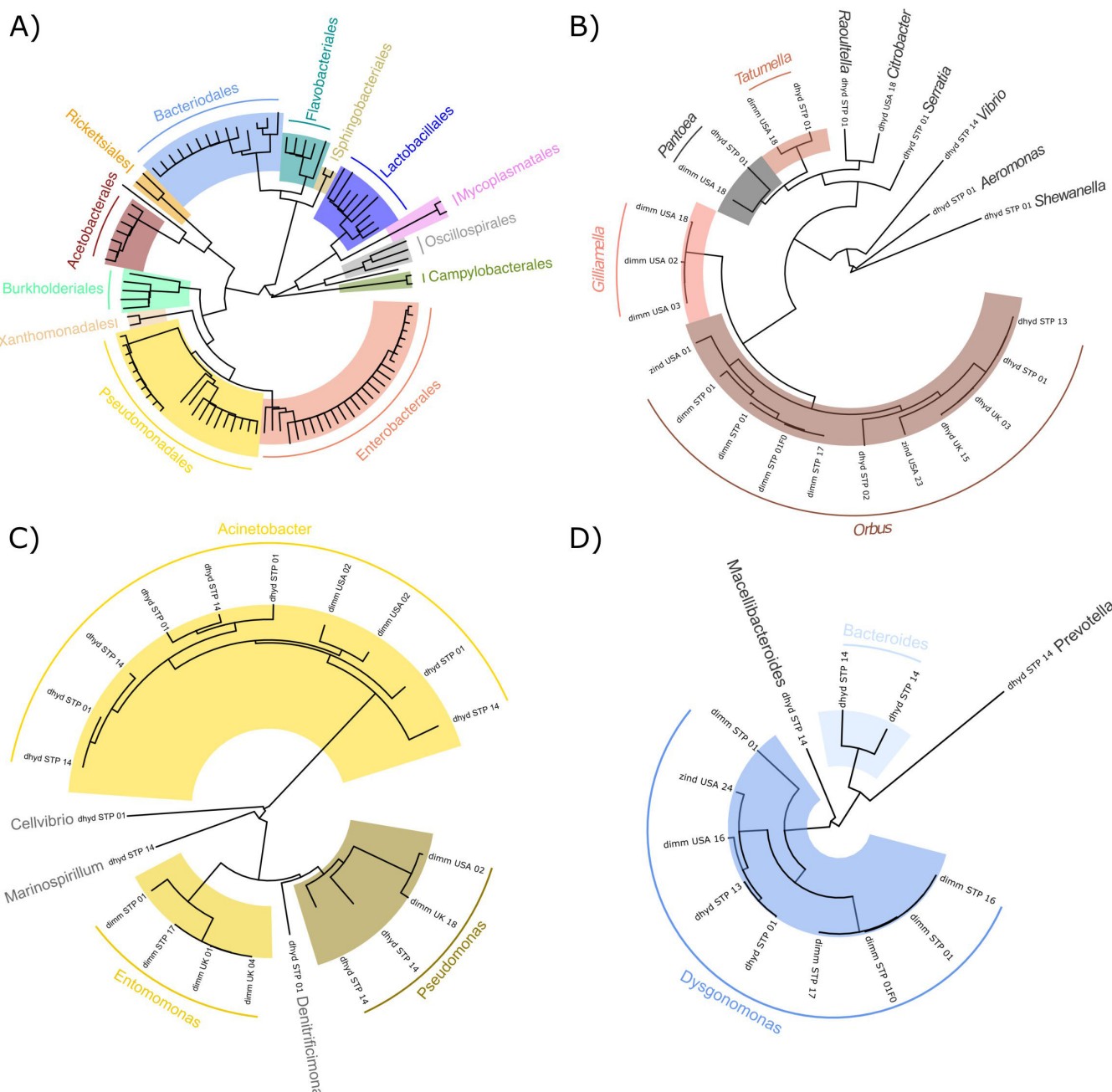

**FIG 2** Phylogenetic classifications of drosophilid-derived MAGs. (A) Phylogenetic relationships among *Drosophila* MAGs from maximum-likelihood placement by pplacer as implemented in GTDB-TK's 'classify_wf' pipeline against the GTDB-Tk reference tree. Phylogenetic relationships among *Drosophila* MAGs from the three most abundant orders—Enterobacterales (B), Pseudomonadales (C), and Bacteriodales (D)—are generated using GTDB-Tk's 'de_novo_wf' pipeline, highlighting genera represented by multiple MAGs within each order. In panels B to D, details on the host species, individual sample identifier, and location are reported in tip names (details as in Fig. 1).

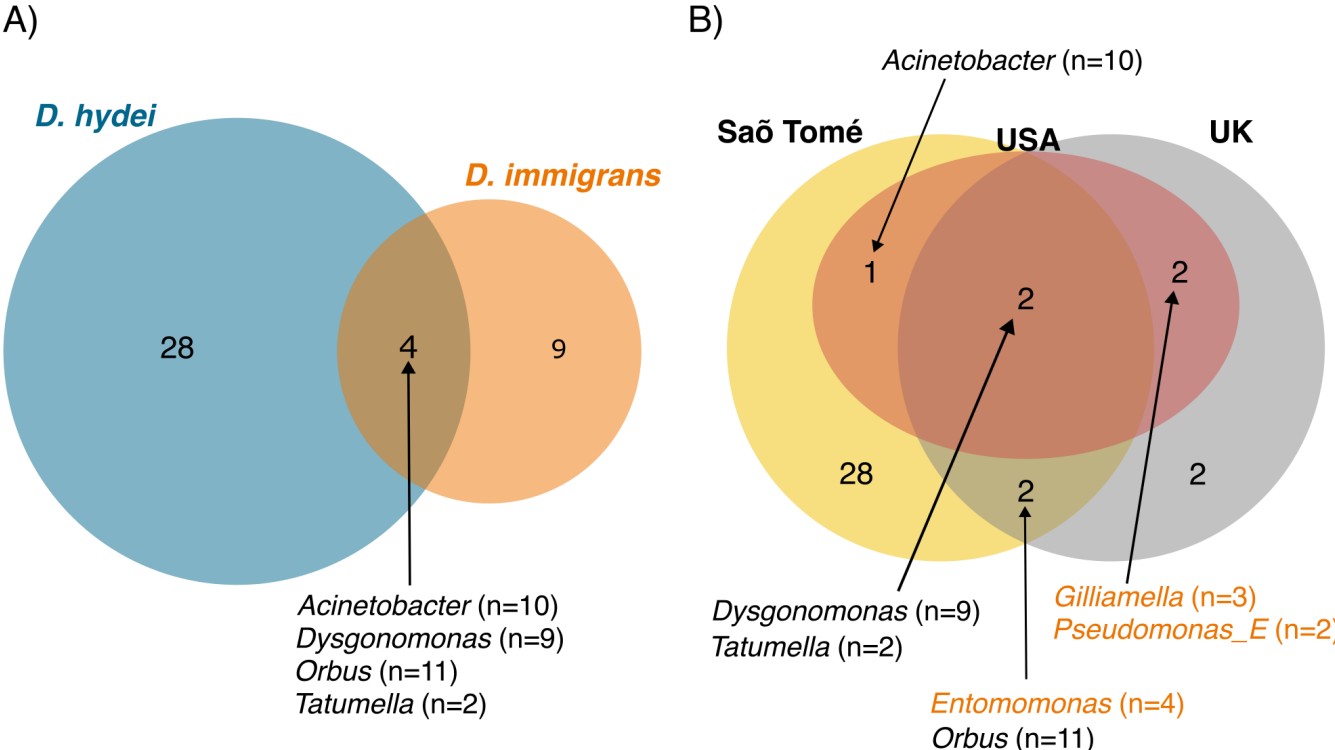

**FIG 3** Core bacterial genera are shared among *Drosophila* species (A) and across geographic regions we sampled (B). Bacterial genera (number inside circles) and the number of MAGs assembled for each genus (beside generic names) are given for each scenario of overlap. Bacterial genera highlighted in orange (*Entomomonas*, *Gilliamella*, and *Pseudomonas_E*) were only assembled from *D. immigrans* hosts. Note that numbers of MAGs belonging to *Dysgonomonas* and *Orbus* include MAGs assembled from *Z. indianus* (see Fig. 2), and *D. repleta* were grouped with *D. hydei* for this analysis.

both from the single F1 female *Z. tsacasi* in our dataset, and six MAGs were classified in the family Lactobacillaceae. Five of the Lactobacillaceae MAGs were derived from flies collected from São Tomé (two *D. repleta* individuals, one *D. immigrans*, and one *Z. tsacasi*), and the sixth from a *Z. indianus* individual collected in North Carolina, USA. We focused functional analyses on 62 MAGs from bacteria belonging to the three orders (and seven genera) assembled from multiple host species and locations. These MAGs are either associated with a common environment used by the species we sampled, or they are evolutionarily associated as core members of the human-commensal drosophilid microbiome.

Endosymbiotic bacteria sequences were also abundant in four of our 31 samples: sequences classified as *Wolbachia* were abundant in *Z. taronus* and *Z. tsacasi* collected from São Tomé (39,265 and 2,675 reads, respectively), and high-quality MAGs classified in the diverse group of *Wolbachia pipientis* were assembled from both samples (CheckM completeness >99% and contamination = 0.00%; Table S2 at https://doi.org/10.5281/zenodo.14173039). Phylogenomic analysis of these two MAGs with GTDB-Tk revealed that they are closely related to *w*Mel-type *W. pipientis* that have been isolated from *Drosophila melanogaster* (RefSeq assembly GCF_000008025.1) and *Diaphorina citra* (GenBank assembly GCA_902636535.1) (see Fig. S8 at https://doi.org/10.5281/zenodo.14173039). Sequences classified as *Spiroplasma*—another endosymbiont found in arthropods—were abundant in two *D. hydei* collected from the UK (86,369 and 76,409 reads), and MAGs classified as *Spiroplasma poulsonii* were assembled from both samples (CheckM completeness = 98.5% and 72.18%, contamination = 3.01% and 0.00%, respectively; Table S2 at https://doi.org/10.5281/zenodo.14173039).

To explore the hypotheses that "focal" bacterial genera are either found in a common environment shared among human commensal drosophilids or are evolutionarily

associated members of the core drosophilid microbiome, we compared MAGs to publicly available bacterial genomes using GTDB-Tk's 'de_novo_wf' pipeline. Drosophilid-derived MAGs classified as *Gilliamella* and *Orbus* were found to be closely related to bacteria belonging to the genera *Gilliamella*, *Orbus*, and *Frischella* that have been assembled from diverse honey bee (*Apis*) and bumble bee (*Bombus*) hosts (Fig. 4A) (27, 29). *Gilliamella* and *Orbus* MAGs derived from the drosophilid hosts we sampled tended to form monophyletic clades relative to bacteria from honey or bumble bees, suggesting host-specific divergence (Fig. 4A). Indeed, *Orbaceae* bacteria have been reported in metagenomic studies of wild mushroom-feeding *Drosophila* (19, 20), and a recent study isolating *Orbaceae* bacteria from wild *Drosophila* described three new species belonging to the genus *Orbus* (30). The *Orbus* MAGs we generated here (see Fig. 2B) are closely

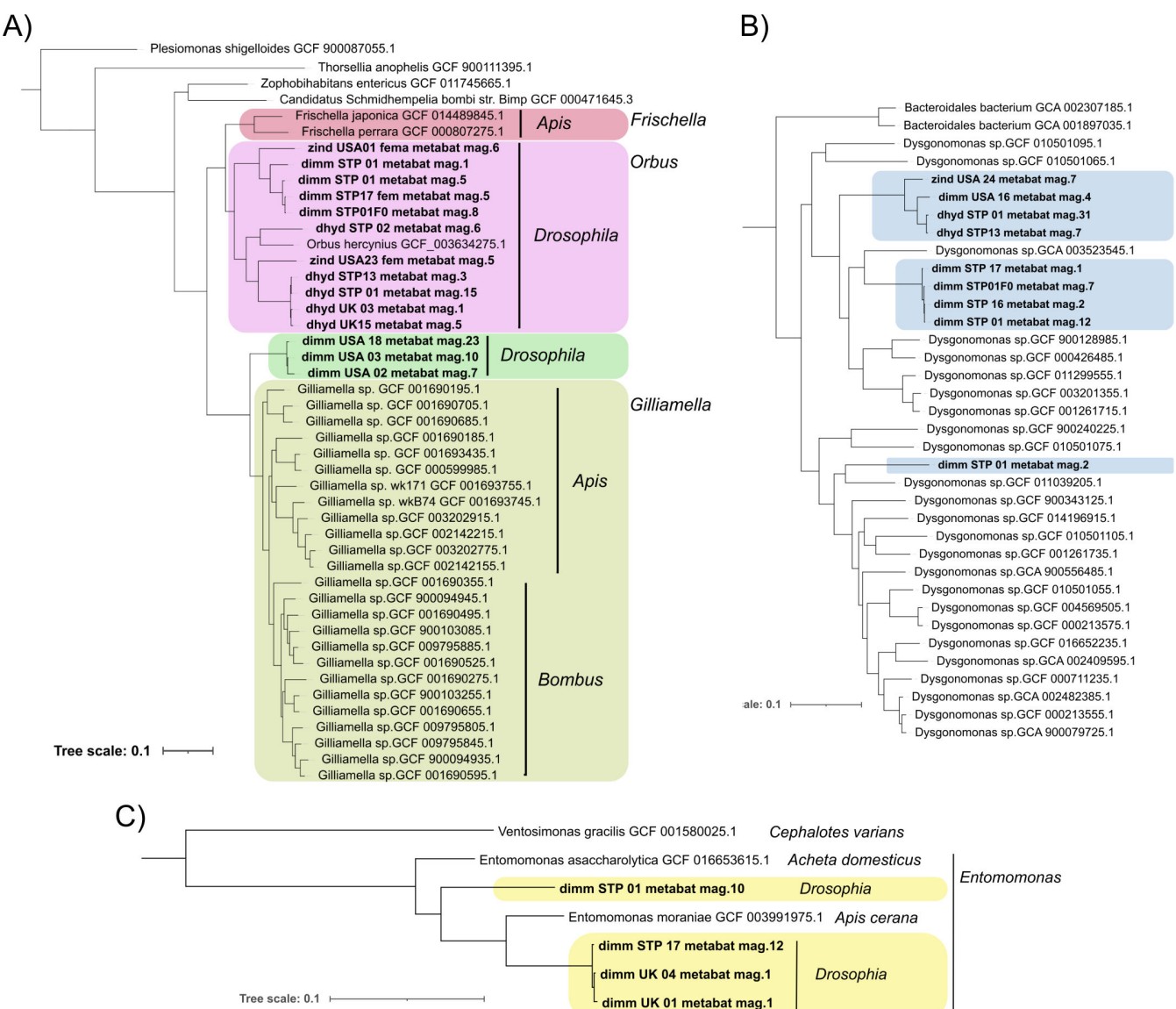

**FIG 4** Phylogenetic relationships among focal MAGs assembled from drosophilid flies (bold type) and publicly available genomes included in GTDB-Tk's 'de_novo_wf' pipeline. (A) Enterobacterale MAGs from drosophilid hosts are closely related to genomes from the genera *Gilliamella*, *Orbus*, and *Frischella* that have been assembled from diverse honey bee (*Apis*) and bumble bee (*Bombus*) hosts. (B) Drosophilid MAGs from the genus *Dysgonomonas* are closely related to diverse *Dysgonomonas* genomes assembled from insect, mammal, and environmental sources (sources not shown in panel B). (C) *Drosophilid* MAGs from the genus *Entomomonas* are closely related to genomes from *Entomomonas* sequenced from the eastern honey bee (*Apis cerana*) and the house cricket (*Acheta domesticus*).

related to *Orbus wheelerorum* and *O. mooreae*, previously described from *Drosophila* spp. hosts (30) (average nucleotide identity >95% as calculated by OrthoANIu [31]; see Fig. S7 at https://doi.org/10.5281/zenodo.14173039), suggesting broad associations between these bacteria and drosophilid hosts.

Drosophilid-derived MAGs classified in the genus *Dysgonomonas* were found to be closely related to *Dysgonomonas* genomes derived from diverse sources, including environmental samples, insects, and mammals (Fig. 4B; see also Table S3 at https://doi.org/10.5281/zenodo.14173039). Drosophilid-derived MAGs classified in the genus *Entomomonas* were related to two *Entomomonas* species in the GTDB-Tk database—one derived from an eastern honey bee (*Apis cerana*) host and the other from a house cricket (*Acheta domesticus*) (Fig. 4C). Phylogenetic comparisons to publicly available genomes therefore suggest that many of the MAGs from the genera *Gilliamella*, *Orbus*, and *Entomomonas* are associated as core members of the insect (gut) microbiome.

## Functional variation among drosophilid-associated bacteria

We quantified enrichment in COG categories and KEGG pathways among MAGs belonging to the seven focal bacterial genera described above to test for evidence of functional differences among them. We found evidence for variation in enrichment of 12 COG categories and 27 KEGG pathways across genes annotated in each MAG (Fig. 5). Comparing the enrichment of KEGG pathways among bacterial genera found that more than 80% of drosophilid-derived *Gilliamella* or *Orbus* genomes are enriched for genes belonging to 10 KEGG pathways that are under-enriched (i.e., fewer than 50% of MAGs with enrichment) in the 35 Pseudomonadales and Bacteriodales MAGs in our dataset (Fig. 5B). *Acinetobacter*, *Entomomonas,* or *Pseudomonas* genomes are enriched for genes belonging to 14 KEGG pathways that are under-enriched in the 43 Enterobacterales and Bacteriodales MAGs in our dataset (Fig. 5C). *Dysgonomonas* genomes are enriched for genes belonging to four KEGG pathways that are under-enriched in the 48 Pseudomonadales and Enterobacterales MAGs in our dataset (Fig. 5D).

KEGG pathways that are enriched in *Gilliamella* or *Orbus* MAGs tend to be enriched across both genera: an average of 87.5% of MAGs in these genera shows enrichment in the same 10 KEGG pathways (range = 58.3% to 100%; Fig. 4B). Seven of the 10 KEGG categories that are enriched in *Gilliamella* and *Orbus* MAGs are also enriched in more than 50% of other Enterobacterales MAGs (map00051, map00052, map00480, map00500, map00564, map01503, and map02060), suggesting that functions associated with these pathways are shared across the Enterobacterales species associated with drosophilid hosts. However, three KEGG pathways (map00040, map00053, and map01501) are enriched in fewer than 50% of other Enterobacterales MAGs, suggesting

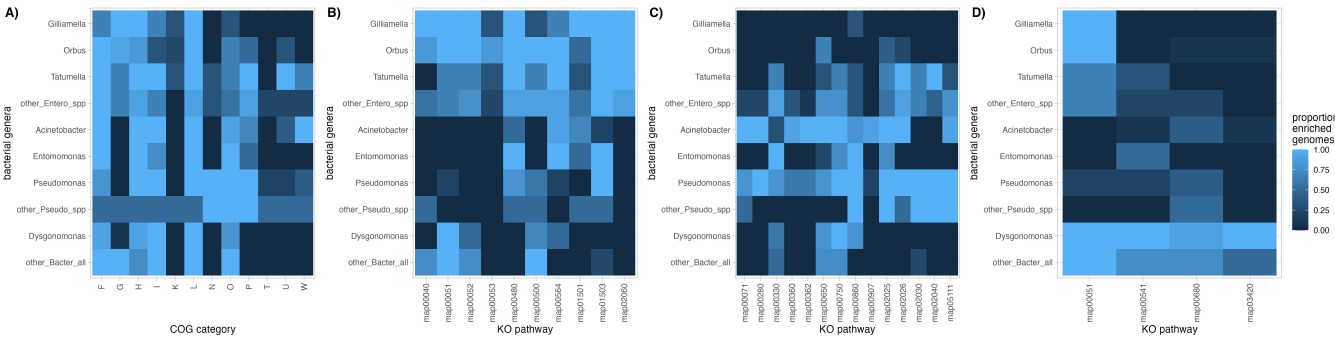

**FIG 5** MAGs belonging to different bacterial genera differ in their functional gene content. (A) COG categories that show variation in enrichment among MAGs. (B) KEGG pathways that are enriched in greater than 80% of the MAGs assembled from *Gilliamella* or *Orbus* (order: Enterobacterales) but were enriched in fewer than 50% of other MAGs in our dataset. (C) As in panel B, but with overrepresented enrichment within genera *Acinetobacter*, *Entomomonas*, or *Pseudomonas* (order: Pseudomonadales). (D) As in panel B, but with overrepresented enrichment within the genus *Dysgonomonas* (order: Bacteriodales). In panel A, the letters on the x-axis are the one-letter identifiers for COG categories. Categories F, G, H, I, and P are involved in "metabolism"; K and L are involved in "information storage and processing"; and N, O, T, U, and W are involved in "cellular processing and signaling."

possible unique or enhanced functions within *Gilliamella* and/or *Orbus*. Two of these three pathways contain genes involved in carbohydrate metabolism—specifically in pentose and glucuronate interconversions (map00040) and ascorbate and aldarate metabolism (map00053)—while the third (map01501) is involved in resistance to beta-lactam antibiotics. Interestingly, map00053 is only enriched in one of the three *Gilliamella* MAGs, while map01501 is enriched in all three *Gilliamella* MAGs and six of the 12 (50%) *Orbus* MAGs, suggesting potential functional differences between strains of these closely related genera.

In contrast to Enterobacterales, there is more variation in enrichment among Pseudomonadales from the genera *Acinetobacter*, *Entomomonas*, and *Pseudomonas*: an average of 62.7% of MAGs from these genera show enrichment in the same KEGG pathway (range = 33.3% to 93.3%; Fig. 5C). Within Bacteriodales, the four KEGG pathways that are enriched across the 10 drosophilid-derived *Dysgonomonas* MAGs tend to be enriched in the other Bacteriodales MAGs (50%–100%); however, we note that there are only four MAGs assigned to the order Bacteriodales that were not within the genus *Dysgonomonas* in our dataset. Nonetheless, all four KEGG pathways enriched in *Dysgonomonas* (map00051, map00541, map00680, and map03420) were rarely enriched in non-Bacteriodales (44.2%, 16.9%, 20.1%, and 2.3%, of MAGs with enrichment, respectively; Fig. 5D). Map00051 was enriched in *Dysgonomonas*, *Gilliamella*, and *Orbus* genomes (Fig. 5D, first column) and was involved in the metabolism of fructose and mannose sugars. The other three pathways contain genes involved in glycan sugar biosynthesis and metabolism, methane metabolism, and nucleotide excision repair. Taken together, variations in KEGG pathway enrichment across *Drosophila*-associated MAGs indicate that bacterial members of the *Drosophila* microbiome are functionally different, for example, via differences in metabolic capacity for various substrates (e.g., carbohydrates versus methane) or the ability to resist environmental stressors (e.g., antimicrobials or general DNA damage).

We also found that *Drosophila*-associated *Gilliamella* and *Orbus* MAGs differed in functional enrichment when compared to publicly available *Gilliamella* genomes isolated from *Apis* and *Bombus* hosts. COG categories N (cell motility) and U (intracellular trafficking, secretion, and vesicular transport) are enriched in *Apis*- and/or *Bombus*-associated genomes, but are not enriched across the majority of *Drosophila*-associated *Gilliamella* or *Orbus* MAGs. However, 4/11 (36.36%) of the *Orbus* MAGs isolated from *Drosophila* were enriched for COG category U. In addition to COG categories, 11 KEGG map pathways varied in enrichment among *Gilliamella* (*Drosophila*), *Orbus* (*Drosophila*), *Gilliamella* (*Apis*), and *Gilliamella* (*Bombus*) MAGs or genomes (Table 1). Four pathways (map00053, map00561, map00561, and map00630) are enriched in *Orbus* genomes derived from *Drosophila* hosts relative to genomes from the other four groups. These pathways contain genes involved in the metabolism of various carbohydrates and lipids

**TABLE 1** KEGG pathways that showed variation in enrichment across genomes from *Orbus* from *Drosophila* hosts, *Gilliamella* from *Drosophila* host, *Gilliamella* from *Apis* host, or *Gilliamella* from *Bombus* host[a]

| KEGG ID | Pathway name | Orbus | *Gill. (Dros.)* | *Gill. (Apis)* | *Gill. (Bombus)* |
|---|---|---|---|---|---|
| map00053 | Ascorbate and aldarate metabolism | *0.82* | 0.33 | 0.25 | 0.08 |
| map00450 | Selenocompound metabolism | *0.82* | *1.00* | 0.50 | 0.00 |
| map00561 | Glycerolipid metabolism | *0.82* | 0.00 | 0.33 | 0.00 |
| map00564 | Glycerophospholipid metabolism | *1.00* | 0.67 | 0.08 | 0.15 |
| map00630 | Glyoxylate and dicarboxylate metabolism | *1.00* | 0.33 | *0.83* | 0.15 |
| map00261 | Monobactam biosynthesis | 0.45 | *1.00* | 0.08 | 0.00 |
| map01501 | Beta-lactam resistance | 0.55 | *1.00* | 0.17 | 0.38 |
| map02025 | Biofilm formation – *Pseudomonas aeruginosa* | 0.45 | 0.00 | 0.42 | *0.85* |
| map02026 | Biofilm formation – *Escherichia coli* | 0.27 | 0.00 | *1.00* | *1.00* |
| map02030 | Bacterial chemotaxis | 0.00 | 0.00 | *1.00* | *1.00* |
| map02040 | Flagellar assembly | 0.00 | 0.00 | *1.00* | *1.00* |

[a]The proportion of genomes of each group that showed enrichment in each pathway is given. See Fig. 4A for relationships among these groups. Values above 0.8 are highlighted in bold italic font.

(Table 1). One of these pathways (map00630) is also enriched in *Gillimella* genomes derived from hosts in the genus *Apis*. Three pathways (map00261, map00450, and map01501) are enriched in *Gilliamella* derived from *Drosophila* hosts: maps 00261 and 01501 are both involved in antimicrobial production, and map01501 was also less likely to be enriched in *Drosophila*-derived MAGs that were not in the genus *Gilliamella* (Fig. 5B). Four pathways involved in biofilm formation or movement (map02025, map02026, map02030, and map02040) are enriched in *Gilliamella* derived from both *Bombus* and *Apis* hosts. The two pathways involved in biofilm formation—map02025 and map02026 —are also enriched in five (45%) and three (27%) of the *Orbus* MAGs, respectively. While the KEGG pathways above highlight potential functional differences among bacteria within the family *Orbaceae* (which includes the genera *Gilliamella*, *Orbus*, and *Frischella*) from different hosts, we also found 51 KEGG pathways that are enriched across all *Orbaceae* genomes (see Table S4 at https://doi.org/10.5281/zenodo.14173039). These pathways highlight that *Orbaceae* also share diverse metabolic and biosynthetic pathways.

## Biosynthetic potential of drosophilid-associated bacteria

Using antiSMASH, we were able to annotate secondary metabolite biosynthetic gene clusters (BGCs) in 56 of the 62 MAGs classified as Enterobacterales, Pseudomonadales, or Bacteroidales (Fig. 6A). Across MAGs, we identified 177 biosynthetic gene clusters from 31 small molecule classes and their hybrids (table in S5). Across the 56 MAGs that carried at least one BGC, we annotated an average of three BGCs per MAG, found more BGCs per MAG in MAGs from the order Pseudomonadales (maximum of 14 BGCs per MAG) (KW $\chi^2 = 14.683$, $P < 0.001$; see Fig. S6A at https://doi.org/10.5281/zenodo.14173039), and found a significant positive relationship between the genome size and the number of BGCs per MAG (ANOVA on genome size; $F = 48.668$, $P = 3.8 \times 10^{-9}$; see Fig. S6B at https://doi.org/10.5281/zenodo.14173039). The most abundant class of BGCs was post-translationally modified peptides (RiPP-like) ($n = 40$), followed by aryl polyene ($n = 28$), non-ribosomal peptides' hybrids (Other NRPS) ($n = 22$), non-ribosomal peptides (NRPS) ($n = 19$), and beta-lactone ($n = 15$) (Fig. 6B). In total, MAGs classified

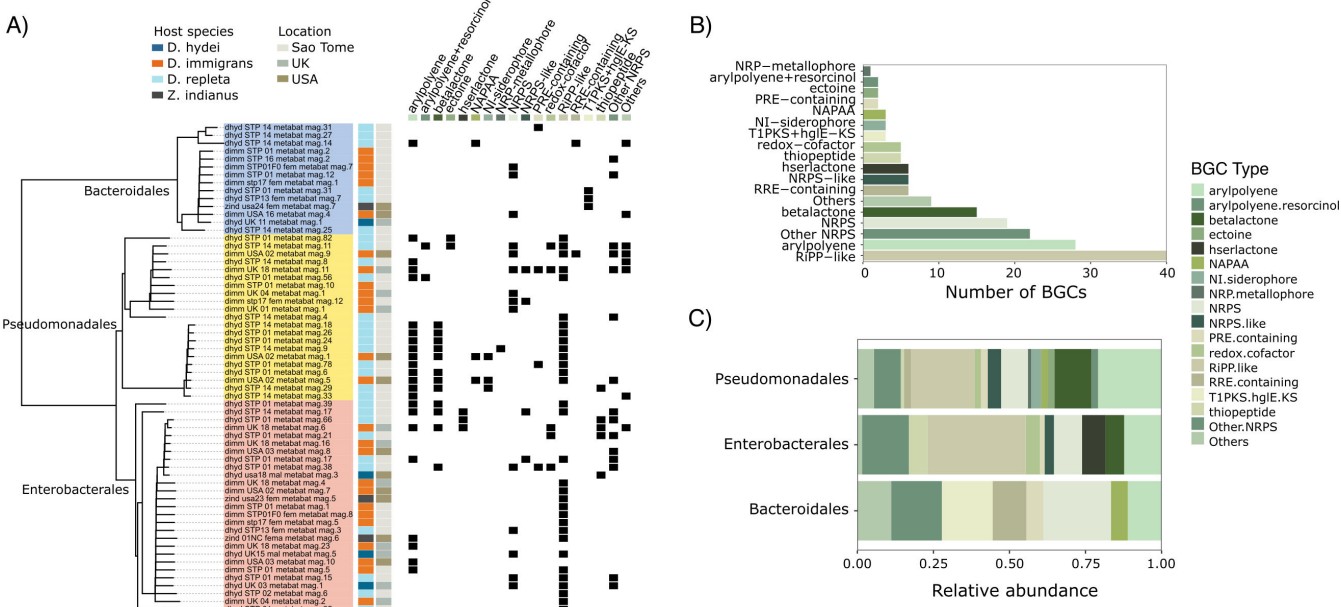

**FIG 6** *Drosophila* MAGs differ in the secondary metabolite biosynthetic gene clusters (BGCs) they possess. (A) Summary of biosynthetic products for MAGs from the Bacteriodales, Pseudomonadales, and Enterobacterales, including details of the host species and location the flies were collected from. Across MAGs, we identified 177 biosynthetic gene products belonging to 18 types of BGC (B). The relative abundance of BGC types was significantly different among the three focal orders of bacteria included in this analysis (C).

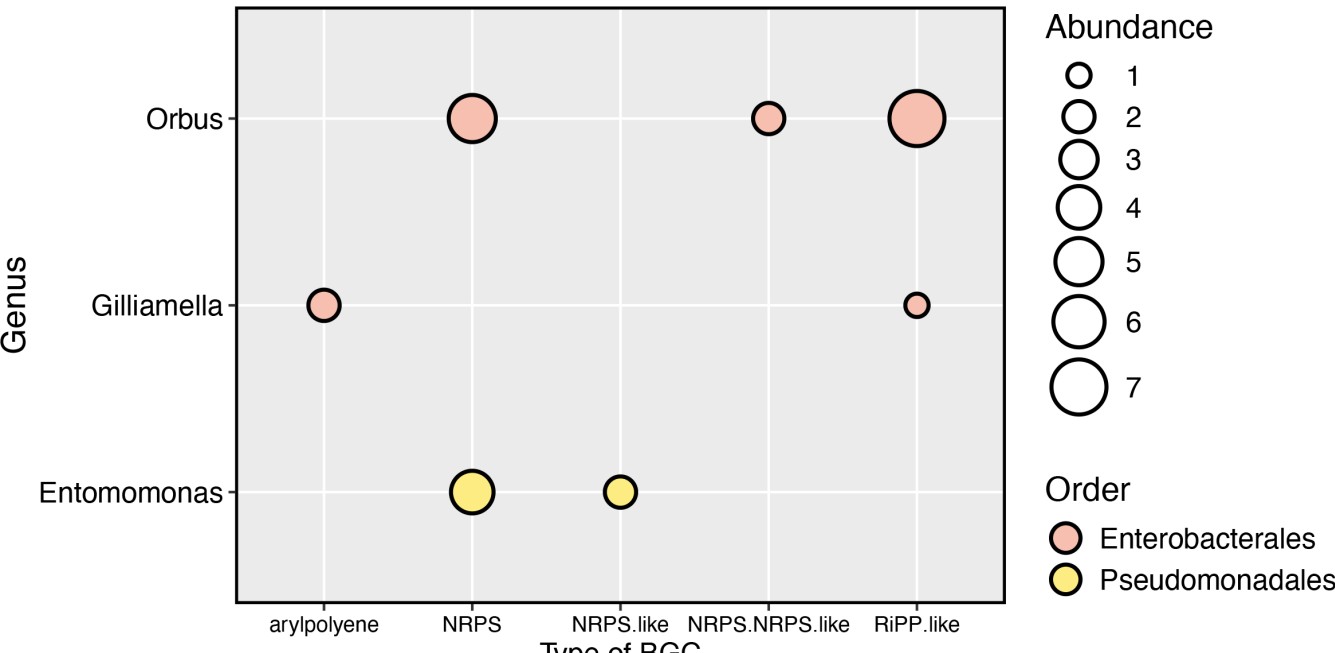

**FIG 7** Abundances of biosynthetic gene cluster (BGC) classes (i.e., "Type of BGC") that were the most common within the focal bacterial genera *Orbus*, *Gilliamella*, and *Entomomonas*.

as Pseudomonadales possessed 91 biosynthetic gene clusters from 24 BGC classes, Enterobacterales possessed 68 biosynthetic gene clusters from 16 BGC classes, and Bacteroidales possessed 18 biosynthetic gene clusters from nine BGC classes. The relative abundance of BGC classes differed among our three focal bacterial orders (Permutational MANOVA: $F = 3.722$; $P = 0.001$; Fig. 6): for example, 32% of BGCs in Enterobacterales were in the class RIPP-like, while 20% were RIPP-like in Pseudomonadales, and RIPP-like BGCs were absent in Bacteroidales. Aryl polyene class made up 20%, 12%, and 11% of BCGs annotated in Pseudomonadales, Enterobacterales, and Bacteroidales MAGs, respectively; NRPS hybrids (Other NRPS) made up 16%, 15%, and 8% of BCGs annotated in Bacteroidales, Enterobacterales, and Pseudomonadales MAGs; and NRPS made up 22%, 9%, and 8% of BGCs annotated in Bacteroidales, Enterobacterales, and Pseudomonadales MAGs. Beta-lactone BCGs were found in Pseudomonadales (20%) and Enterobacterales (12%) but were absent in Bacteroidales. The only BGC class that was unique to Bacteroidales was hybrid T1PKS + hglE KS (16%) (Fig. 6C). MAGs from the genera *Gilliamella*, *Orbus*, and *Entomomonas* possessed 23 biosynthetic gene clusters from four BGC classes: aryl polyene ($n = 2$), NRPS ($n = 9$), NRPS-like ($n = 2$), NRPS+NRPS-like hybrid ($n = 2$), and RiPP-like ($n = 8$). The most abundant class of BGCs in *Gilliamella*, *Orbus*, or *Entomomonas* MAGs were aryl polyene ($n = 2$), RiPP-like ($n = 7$), and NRPS ($n = 5$), respectively (Fig. 7). Taken together, variation in the predicted BGCs among drosophilid-associated bacteria suggests that these bacteria can produce a diverse range of secondary biosynthetic molecules, and bacteria in different orders vary in the secondary metabolites they produce.

## DISCUSSION

Insect-associated bacteria can have diverse effects on their host's biology—for example, they can detoxify dietary toxins (32) and provide protection from infection (7). However, there is also evidence that experimental manipulation of insect microbiomes (via antibiotic treatment) does not affect growth and development (33). These conflicting results highlight a need to identify the specific members of host-associated microbial communities and their functions. Consistent with previous studies of the microbiomes

of wild drosophilids (10, 19, 20), we recovered a diverse set of host-associated bacterial reads from whole-organism sequencing (Fig. 1). By assembling and analyzing MAGs from these reads, we found that the species of *Drosophila* (including *Zaprionus*) we study here are host to diverse lineages of bacteria (Fig. 2 to 4) that differ in their functional gene content (Fig. 5 to 7).

Bacteria in the family *Orbaceae* were among the most abundant taxa across samples, and we assembled MAGs from this family from multiple host species and locations (Fig. 2 to 4). *Orbaceae* have previously been reported in metabarcoding studies of wild *Drosophila* as dominant members of their microbiome (19, 20, 34) and have also been reported from *Apis*, *Bombus*, and *Xylocopa* bees (25–27, 29, 35), *Eristalis* flies (36), and *Heliconius* and *Sasakia* butterflies (37, 38). The MAGs we report as *Orbaceae* in this study were assembled from the host species *Z. indianus*, *D. immigrans*, *D. hydei*, and *D. repleta* and from locations in the USA, the UK, and São Tomé (Fig. 4A; also see Fig. S7 at https://doi.org/10.5281/zenodo.14173039). The MAGs we classified as belonging to *Orbacea* that were assembled from *D. repleta* and *D. hydei* hosts also showed high average nucleotide identity (>95%) to published sequences from *Orbus mooreae* and *O. wheelerorum* that were isolated from Drosophilid flies collected in Texas (30). *Orbaceae* MAGs assembled from *D. immigrans* and *Z. indianus* comprise monophyletic clades that are distinct, yet closely related to *Orbus* and *Gilliamella* spp. derived from other insect hosts (Fig. 4A; also see Fig. S7 at https://doi.org/10.5281/zenodo.14173039). The diverse insect hosts, along with the widespread geographic sampling of *Orbaceae*, indicates that they are likely "core" members of many insect's microbiomes. *Gilliamella apicola* is a species from the family *Orbaceae* found in bee hosts that has been shown to also have functional metabolic capacities that complement other members of the honey bee gut microbiome (27). For example, *G. apicola* can detoxify toxic sugars found in the diets of bees (28) and can produce biosynthetic molecules that protect bees from infection by the bacterial pathogen *Melissococcus plutonius* (7). These findings highlight the important functional roles that *Orbaceae* bacteria can play in their insect hosts; however, tests that confirm the functional roles of the *Orbaceae* across diverse hosts are still needed.

Bottom-up genomic approaches that characterize the genes carried by a species' genome can help point to the functional potential of those species. Using functional annotations, we found that *Drosophila*-associated *Orbaceae* MAGs are enriched for genes involved in pentose and glucuronate interconversions (map00040), ascorbate and aldarate metabolism (map00053), and resistance to beta-lactam antibiotics (map01501) (Fig. 4). Drosophilid-associated *Orbaceae* MAGs also harbored secondary metabolite biosynthetic gene clusters (BGCs) in the aryl polyene and RiPP-like classes. Ribosomally synthesized and post-translationally modified peptides (RiPP) have diverse roles, including playing a role in microbial interactions and antimicrobial activity (39, 40), and effects of RiPP-like BGCs on pathogens have been described from *Gilliamella* strains isolated from bees (7). The diverse gene products produced by these bacteria highlight how insects and their microbiomes may be an important source of novel bioactive molecule discovery (41). While functional validation is needed—along with quantifying the impacts that drosophilid-associated *Orbaceae* have on host performance and fitness—, the pathways and BSCs that are enriched in the drosophilid-associated MAGs we analyze here are candidates that could inform future functional studies. Indeed, a recently developed Pathfinder plasmid system has been verified in *Orbaceae* bacteria (42) and could be used to facilitate tests of candidate genes and pathways in *Orbaceae* isolated from diverse hosts.

In addition to bacteria from the family *Orbaceae*, we assembled four MAGs from the genus *Entomomonas* from *D. immigrans* collected in the USA and São Tomé and Príncipe (Fig. 4C), as well as MAGs from the genus *Dysgonomonas* from *D. immigrans*, *D. repleta*, and *Z. indianus* collected in the USA and São Tomé and Príncipe (Fig. 4B). Phylogenomic comparisons of drosophilid-associated *Entomomonas* MAGs to publicly available bacterial genomes showed that they are related to *Entomomonas* strains isolated from different insect orders (Fig. 4C). By contrast, publicly available genomes

from the genus *Dysgonomonas* are derived from both host-associated and environmental sources (Fig. 4); however, *Dysgonomonas* has been reported as one of the core members of the microbiome of wild cactophilid *Drosophila* (34). By comparing relationships among drosophilid-associated MAGs and publicly available microbial genomes, our analyses suggest that *Orbaceae*, *Entomomonas*, and (to a lesser extent) *Dysgonomonas* are bacterial genera that may have evolved to utilize insects as hosts. As data from insect-associated microbial communities increase, identifying these "core" members of the insect microbiome will facilitate tests of the assembly and functional rules governing bacteria–insect interactions—for example, testing whether they represent generalist interactions or tightly coevolved symbioses (43).

We recovered a notably low frequency of *Lactobacillus* and *Acinetobacter* across our samples (0.22 and 0.18% median percent of reads across samples, respectively). A large portion of the research on *Drosophila* spp. microbiomes has focused on *Drosophila melanogaster* and the symbiont *Lactobacillus plantarum* (44, 45), which has in turn become a model to understand the genetic underpinnings of facultative symbiosis because the bacteria can affect growth and fertility (46–48). One possibility for the general lack of *Lactobacillus* across our samples is that it is mostly associated with *D. melanogaster* and that the evolution of the microbiome composition differs between species. This would be a corollary from the observed effect of host genotype in microbiome variation in *D. melanogaster* (49). A second possibility is that *Lactobacillus* increases in frequency in *Drosophila* lines that are maintained in laboratory conditions. Sampling of wild North American *D. melanogaster* indicates that the association between this species and *Lactobacillus* indeed occurs in nature (50). Altogether, these results indicate that, not surprisingly, the microbiome of different species of *Drosophila* differs and that further studies to characterize the evolutionary and ecological factors that affect the microbiome (e.g., tropicality and diet) across the drosophilid family are needed.

We annotated an average of three BCGs for each drosophilid-associated MAG belonging to the Enterobacterales, Pseudomonadales, and Bacteriodales, indicating that members of the drosophilid microbiome have the capacity to produce potentially important secondary metabolites, despite generally not having large genomes (see Fig. S6 at https://doi.org/10.5281/zenodo.14173039). Secondary metabolites can play important roles in host–microbiome interactions and have previously been described in members of the gut microbiomes of honey and bumble bees (7), herbivorous turtle ants (51), and mosquitoes (52). Secondary metabolites produced by BGCs are also involved in the regulation of symbiosis in fungus-farming termites (53), the pathogenicity of malaria mosquitoes (52), and the deoxytication of β-methylamino-L-alanine in cycad-feeding insects (54). BGCs belonging to the RiPP class were the most abundant in our drosophilid-associated MAGs and may play a particularly important role in the microbiome since it has been observed that they can function to inhibit the growth of pathogens in bees (7) and can be involved in microbiome–host communication (55). Likewise, aryl polyenes (the second most abundant class in our dataset) have been observed to function as antioxidants, preventing stress caused by reactive oxygen species produced by the host in the case of bees (7, 56). Whether these molecules perform similar functions in drosophilid hosts remains to be confirmed; however, our results contribute to a growing body of work suggesting that BGCs possessed by the gut microbiome bacteria of insects contribute to the hosts' biology and are a rich source of secondary metabolites (7, 57).

*Wolbachia* and *Spiroplasma* are well-known endosymbionts in insects, including in *Drosophila* (58–60), and we recovered sequences from these genera in four of our 31 samples. From these sequences, we were able to assemble MAGs identified as *Wolbachia* from both *Z. tsacasi* and *Z. taronus* hosts. Both *Z. tsacasi* and *Z. taronus* are forest-dwelling species that we collected on the island of Saõ Tomé, and phylogenetic analyses show that their *Wolbachia* endosymbionts are closely related to *w*Mel-type *W. pipientis* (see Fig. S8 at https://doi.org/10.5281/zenodo.14173040). Recent analyses of the genomes of *Wolbachia* isolated from both *Z. tsacasi* and *Z. taronus* have shown that they contain

the cytoplasmic incompatibility causing operons (*cif*s) *cifA* and *cifB* (61). We confirmed that the two *Wolbachia* MAGs we assembled here also contain *cifA* and *cifB* using BLAST (tblastn; results not shown). Novel *Wolbachia* isolates may be useful in the applications of biocontrol; however, their ability to cause cytoplasmic incompatibility remains to be tested (61). *Wolbachia* infection frequencies have also been shown to be highly variable among populations, species, and geographic regions (58, 62–65), and our results suggest that they are rare or absent across human-commensal drosophilids. Similarly, a screen of *Spiroplasma* in 35 species of *Drosophila* found that only three species—all from the "*repleta*" species group—were host to *Spiroplasma* infections (64). *Spiroplasma* infection rates in *D. hydei* (a member of the *repleta* species group) from the UK have been shown to vary from 15% to 29% across a 9-year period (66). The fact that we only found *Spiroplasma* in two samples of *D. hydei* from the UK is consistent with these past studies and may be indicative of a phylogenetic (and/or geographic) signal of *Spiroplasma* infection in *Drosophila* hosts. We are unable to estimate infection frequencies because we did not sample extensively in any one location. However, our data could prove useful for future phylogenetic or comparative genomic studies, and they provide novel MAGs from both *Spiroplasma* and *Wolbachia*.

From a methodological perspective, we have shown that mining whole-organism sequences generated using long-read ONT sequencers can be a productive approach used to estimate diversity and differences among host-associated microbial communities (Fig. 1). "Mined" bacterial sequences can also be assembled into high-quality MAGs when sufficient bacterial sequence is extracted from the total sequence pool (see Fig. S2 and Table S2 at https://doi.org/10.5281/zenodo.14173039). A limitation of this approach is that the proportion of bacterial sequences recovered from a given sample can vary significantly: we recovered mean and median proportions of bacterial sequence per sample of 0.15 and 0.11, respectively, with a range of 0.03 to 0.62 across samples (see Table S1 at https://doi.org/10.5281/zenodo.14173039). Our rarefaction analysis of taxonomic diversity captured by these recovered sequences showed that samples that have a low proportion (and absolute amount) of recovered bacterial sequences do not reach taxonomic saturation (see Fig. S3 at https://doi.org/10.5281/zenodo.14173039). The amount of bacterial sequence we recovered was also positively correlated with the number of MAGs assembled across our 31 samples (see Fig. S5 at https://doi.org/10.5281/zenodo.14173039). These aspects of our dataset highlight potential limitations of mined metagenomic data to comprehensively characterize variation in the microbiome and conduct community metagenomics. Rather than use a community-level metagenomics approach, comparing high-quality MAGs allows for the identification of potentially important core members of the wild drosophilid microbiome and the functional gene content of those taxa. While this approach is different from metagenomic approaches that quantify gene abundance at the community level, it is well suited to identify and estimate functional variation across specific microbial taxa. Future studies could leverage targeted organ-specific sequencing (e.g., the digestive tract) of a larger number of samples to characterize the frequency of genes across different species and geographic locations.

Mining whole-organism sequences to characterize host-associated microbes is likely to be a particularly useful approach when studying organisms where it is challenging to separate microbes from the host, or when microbial communities change when the organism is raised under artificial conditions. However, this approach is limited in not knowing where on the host the microorganisms are located. In many cases, the location or life history of the microorganisms can be reasonably inferred from knowledge of closely related taxa (e.g., endosymbionts and gut commensals); however, this information should be confirmed with additional species-specific data. Mining microbial sequences from large datasets could be used to extract information from sequencing projects where characterizing the microbial community is not a primary goal. For example, recent work has used whole-organism sequencing of individual *Drosophila* spp. to generate genomic and phylogenetic resources for the group (67). This dataset

includes wild-caught individuals whose data could be mined to quantify host-associated microbial diversity. Because laboratory and wild *Drosophila* show significant differences in the microbial communities they host (10, 19), accurate and transparent metadata need to be published alongside whole-organism sequencing to facilitate meaningful comparisons among host individuals. Obtaining metagenomic data from wild flies across the globe should allow us to revisit classical hypotheses in evolutionary genetics pertaining to the effects of latitude, geographic range, and population connectivity on the genetic diversity encompassed in the microbiome. It will also allow community phylogenetic approaches, such as tests for phylosymbiosis (68, 69), to integrate functional predictions based on genomic information to determine how microbiomes assemble and evolve. A comprehensive characterization of interspecific differences in the microbiome will also allow for a dissection of the importance of the microbiome in animal behavior (70). Our work highlights a need for functional studies of diverse insect microbiomes to gain a holistic view of the diversity, evolutionary history, and functional roles that insect-associated microorganisms play in their diverse hosts and the ecosystems they inhabit.

## MATERIALS AND METHODS

### Specimen collection

We sampled populations of drosophilid flies from the United Kingdom, United States of America, and the island of São Tomé (São Tomé and Príncipe) (see Table S1 at https://doi.org/10.5281/zenodo.14173040). Four of the six host species we sampled—*D. immigrans*, *D. hydei*, *D. repleta*, and *Zaprionus indianus*—are geographically widespread and found in association with "anthropogenic" environments (sometimes referred to as "human commensals"), such as agricultural fields and compost heaps. We included two other species—*Z. taronus* and *Z. tsacasi*—that are forest dwellers found in sub-Saharan Africa. We note that the genus *Drosophila* is paraphyletic, with *Zaprionus* nested within *Drosophila* (24); as such, for brevity, we include the genus *Zaprionus* when referring to "*Drosophila*" samples and bacteria throughout. Individual flies were attracted to banana traps and then collected within 12 hours via aspiration or sweep netting. Flies were then briefly anesthetized with FlyNap (Carolina Biological, USA) and identified under a microscope using diagnostic traits described in "The Encyclopedia of North American Drosophilids" (71) and Yassin and David (72) (72). Collected flies were processed in the field by placing all males and a subset of females in 100% ethanol. At some sample locations, a subset of females was maintained on instant fly media to establish isofemale lines. One of the *D. immigrans* and the *Z. tsacasi* individuals included in our dataset was F1 offspring of those wild females. In total, we generated sequence data from 31 individual flies. Most of these individuals belonged to the "human commensal" species *D. hydei* ($N = 8$), *D. repleta* ($N = 4$), *D. immigrans* ($N = 13$), and *Zaprionus indianus* ($N = 4$) collected from sites in the USA, the UK, and São Tomé and Principe (see Table S1 at https://doi.org/10.5281/zenodo.14173040); however, for comparison, we also included one individual from each of the forest specialists *Z. taronus* and *Z. tsacasi* from São Tomé and Principe. Because we sequenced the entire flies (see below), we also assembled the genome of each sequenced fly, annotated dipteran BUSCO genes, and used alignments of BUSCO genes against published drosophilid genomes (67) to confirm the species identity of each individual (results not shown).

### DNA isolation and sequencing

DNA was isolated from individual flies using a phenol–chloroform protocol developed for obtaining high molecular weight DNA from drosophilid flies for sequencing on ONT sequencers (21). Prior to DNA isolation, tissues were hydrated in hydration buffer and homogenized with sterilized pestles in tissue lysis buffer (see dx.doi.org/10.17504/protocols.io.dm6gpbdn8lzp/v2). Sequence libraries were prepared from individual HMW

extractions following a modified Oxford Nanopore Ligation Sequencing Kit protocol (dx.doi.org/10.17504/protocols.io.dm6gpbdn8lzp/v2) and LSK-110 kits. We sequenced each library on individual R9.4.1 flow cells run on Oxford Nanopore MinION MK1C machines and base-called raw reads (in fast5 format) with Guppy (v6.3.8), specifying the "super high accuracy" model using the option "–config dna_r9.4.1_450bps_sup.cfg" and default quality filtering ($q$-score >10). All base-called reads that passed default quality filtering were then used in subsequent analyses.

## Bacterial diversity across samples

To summarize bacterial diversity, we classified sequences with Kraken 2 run using default parameters (v2.1.2) (73) against Kraken's standard database containing RefSeq archaea, bacteria, viral, plasmid, human, and UniVec_Core sequences. We then generated BIOM-format taxonomic summaries for each sample using the kraken-biom tool (74). Combined BIOM tables were imported and converted into a phyloseq object in R using import_biom and phyloseq functions from the *phyloseq* library (75). We removed sequences not assigned to bacteria and bacterial taxa with low variance across samples (threshold = $1 \times 10^{-7}$). To identify the most abundant bacteria within each sample, we first visualized the variation in the proportion of reads assigned to bacterial classes. To visualize bacterial diversity within a sample, we generated rarefaction curves using the rarecurve function from the *vegan* R library run with a step size of 500. To summarize the differences in the relative abundances of bacteria across samples, we converted the phyloseq object to a DGElist using the 'phyloseq_to_edgeR' function from the *PathoStat* library (76) and conducted a principal coordinate analysis (PCoA) using the plotMDS function from the *limma* R package run with the "gene.selection" option set to "pairwise." Because our sequencing resulted in uneven sequencing depths across samples, we also conducted a redundancy analysis (RDA) using the RDA function from the *vegan* R library to explore how the number of bacterial sequences recovered from an individual fly and the species identity of that fly affected bacterial community composition/abundance. When conducting the RDA, we modeled taxonomic abundance (read counts per bacterial taxon) as a function of the number of bacterial reads recovered from a sample, the host species, and the interaction between the number of bacterial reads and the host species (fixed effects). We tested for significant effects of these terms on taxonomic abundances using the ANOVA function in R.

## Metagenome assembly and classification

Using read-level classifications generated by Kraken2, we identified and isolated bacterial reads from whole-organism fastqs by selecting reads that were not unclassified or classified as "Homo" using seqtk's 'subseq' command (v1.3-r106; https://github.com/lh3/seqtk). We then assembled metagenomic contigs from the pool of putatively bacterial reads with metaFlye (v2.9) (77) run using the "—nano-hq" flag and polished assembled contigs with medaka_consensus (v1.7.2) (78) run using the r941_min_sup_g507 model. Polished contigs were binned into MAGs using metaBAT2 (79), and levels of completeness and contamination were assessed using CheckM ('lineage_wf'; v1.1.3) (80).

We used GTDB-Tk's 'classify_wf' pipeline (81) to assign taxonomy and determine phylogenetic relationships among MAGs with completeness greater than 45% and contamination less than 10%, as determined by CheckM. GTDB-TK leverages the Genome Taxonomy Database (82) and tools that allow for taxonomic assignment through sequence clustering, alignment, and large-scale phylogenetic reconstruction (83–88). To explore the relationships among drosophilid MAGs and publicly available bacterial genomes, we also carried out focused analyses on bacterial genera that were represented by MAGs independently assembled from multiple host individuals in our dataset (see Results) using GTDB-Tk's 'de_novo_wf' pipeline. Similar to classify_wf, the de_novo_wf pipeline uses Prodigal (84) and HMMER (83) to identify marker genes in each MAG. Genes are then concatenated, and phylogenetic relationships are inferred using FastTree (87)

run with the WAG + GAMMA model. These analyses applied taxonomic filters at the order level and required a minimum of 50% of amino acids for a given MAG to be included in the alignment for that MAG to be retained in the analysis. We subsequently implemented de_novo_wf analyses separately for the orders Enterobacterales, Bacteriodales, and Pseudomonodales (outgroups: Pseudomonadales, Sphingobacteriales, and Enterobacterales, respectively). Finally, we manually pruned phylogenetic trees produced by this analysis and compared the host organisms of publicly available bacterial genomes that were within the same genera as focal drosophilid MAGs.

## Annotation and functional characterization of MAGs

We annotated MAGs that received CheckM completeness and contamination scores greater than 45% and 10%, respectively, using eggNOG-mapper v2 (89, 90), run with eggNOG v5.0 (91) and protein predictions made by Prodigal (84). We note that while we used cutoffs of 45% completeness and 10% contamination to identify "focal" MAGs for subsequent analyses, 58% of these MAGs had completeness scores above 90% (82.5% with greater than 70% completeness scores), and the average contamination across these MAGs was 1.6% (see Table S2 at https://doi.org/10.5281/zenodo.14173040). We focused functional comparisons on lineages of bacteria for which we had multiple MAGs assembled from either different drosophilid species or different locations within the same species. Specifically, we initially focused on MAGs assembled for bacterial taxa within the orders Enterobacterales ($N$ = 27 MAGs), Pseudomonodales ($N$ = 21), and Bacteroidales ($N$ = 14), as species within these orders are likely to be core (or symbiotic) members of the microbiomes of the *Drosophila* species we sampled (see Results). For gene sets annotated for each MAG within these three orders, we identified clusters of orthologous gene (COG) categories and KEGG Orthology (KO) pathway maps that were enriched at the genome level using COG and KEGG annotations generated by eggNOG-mapper and the 'enrichCOG' and 'enrichKO' functions in the *MicrobiomeProfiler* R library (v1.6.1) (92).

We compared functional diversity among groups of *Drosophila*-associated bacteria by identifying COG categories and KEGG pathway maps that were enriched in at least 80% of the focal group being assessed and in less than 50% of all other MAGs in our dataset. We tested for functional enrichment among MAGs from each of the genera *Gilliamella*, *Orbus*, *Acinetobacter*, *Entomomonas*, *Pseudomonas*, and *Dysgonomonas*. We also compared enrichment in the *Gilliamella* and *Orbus* MAGs we assembled as part of this study with enrichment in 12 publicly available *Gilliamella* genomes derived from *Apis* bees and 13 derived from *Bombus* bees (*Gilliamella* clade in Fig. 4A; see also Table S3 at https://doi.org/10.5281/zenodo.14173039).

Finally, we predicted biosynthetic gene products produced by MAGs classified in the orders Enterobacterales, Pseudomonodales, or Bacteroidales using the "antibiotics and secondary metabolite analysis shell" (antiSMASH v7) (93). We first annotated genes in each MAG belonging to these three families using Prokka (v1.14.5) run with default parameters. Prokka annotations were then used as input to antiSMASH to identify biosynthetic gene clusters (BGCs) within each MAG. We ran the online version of antiSMASH (https://antismash.secondarymetabolites.org/) (93) with relaxed detection strictness and all options on. For visualization and analyses, we labeled singleton biosynthetic classes as "Others" and non-ribosomal peptides' hybrid classes as "Other NRPS". We performed Kruskal–Wallis (KW) and post-hoc Wilcoxon ranks sum test to determine differences in the frequency of BGCs per MAG between bacteria orders. We fit a linear model to test for effects of genome size and bacterial order on the ($\log_{10}$) number of BGCs per MAG. In this analysis, we first tested for an interaction between genome size and bacterial order, but this was not significant. We therefore include genome size and bacterial order as predictors in the final model, but not the interaction between the two. Finally, to test whether the presence/abundance of BGC classes varied across bacterial orders, we conducted a permutational analysis of variance (PERMANOVA)

on Euclidean distances between MAGs as implemented with the adonis function in the vegan package in R (94).

## ACKNOWLEDGMENTS

We thank Darren Obbard for sharing flies collected in the UK, Brandon Cooper for assistance with collecting flies in São Tomé, and two anonymous reviewers for their constructive comments on a previous draft of the manuscript.

This work was supported by a Royal Society Research Grant, the Royal Society of London (RGS\R1\221323 awarded to A.A.C.), and the NERC Envision Doctoral Training Program (NE/L002604/1 awarded to A.H.O.). D.R.M. is funded by the National Institute of General Medical Sciences under award R35GM148244.

## AUTHOR AFFILIATIONS

[1]School of Environmental and Natural Sciences, Molecular Ecology & Evolution Group, Prifysgol Bangor University, Bangor, United Kingdom
[2]Department of Biology, University of North Carolina, Chapel Hill, North Carolina, USA

## AUTHOR ORCIDs

Aaron A. Comeault  http://orcid.org/0000-0003-3954-2416
Daniel R. Matute  http://orcid.org/0000-0002-7597-602X

## FUNDING

| Funder | Grant(s) | Author(s) |
| --- | --- | --- |
| Royal Society | RGS\R1\221323 | Aaron A. Comeault |
| National Institute of General Medical Sciences | R35GM148244 | Daniel R. Matute |

## DATA AVAILABILITY

Raw sequence reads are available in the NCBI SRA under BioProject no. PRJNA1188364. Supporting figures, tables, scripts, and MAGs are available at Zenodo: doi:10.5281/zenodo.14173039.

## ADDITIONAL FILES

The following material is available online.

### Open Peer Review

**PEER REVIEW HISTORY (review-history.pdf).** An accounting of the reviewer comments and feedback.

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
