## [Reviewer comments · mSystems]

Phylogenetic and functional diversity among *Drosophila*-associated metagenome-assembled genomes

Aaron Comeault, Alberto Orta, David Fidler, Tobias Nunn, Amy Ellison, Tayte Anspach, and Daniel Matute

Corresponding Author(s): Aaron Comeault, Bangor University

Review Timeline:

Submission Date:	January 22, 2025
Editorial Decision:	February 13, 2025
Revision Received:	April 30, 2025
Accepted:	June 8, 2025

Editor: Laetitia Wilkins

Reviewer(s): Disclosure of reviewer identity is with reference to reviewer comments included in decision letter(s). The following individuals involved in review of your submission have agreed to reveal their identity: Irene L. G. Newton (Reviewer #2)

Transaction Report:

DOI: <https://doi.org/10.1128/msystems.00027-25>

Re: mSystems00027-25 (Phylogenetic and functional diversity among Drosophila-associated metagenome-assembled genomes)

Dear Dr. Aaron A Comeault:

Revision Guidelines

Sincerely,
Laetitia Wilkins
Editor
mSystems

Reviewer #1 (Comments for the Author):

This study by Comeault et al. about metagenomics of flies in the family Drosophilidae. They collected wild caught and F1 flies from three countries, UK, USA, Sao Tome and Principe. The study is well-designed with thorough analysis. However, the methods section, especially the statistical analyses, needs better description. The discussion could also be expanded.

Introduction

L36: "microbiomes of non-model organisms"?

I think the last paragraph of the introduction is somewhat too detailed. Some information can be put in the methods and results sections.

Methods

The authors lacked sufficient detail in their analysis. Parameters of several programs used were not specified (e.g., default parameters,..).

Please provide more information on the collection sites such as coordinates in Table S1. For example, North Carolina is not so informative.

Can you please provide the key used for fly identification.

Why did the authors include *Z. taronus* and *Z. tsacasi* in this study because there is only one sample per species.

L137: did the authors follow the extraction protocol exactly as described? Also the authors should cite this paper instead <https://doi.org/10.7554/eLife.66405> .

How did you do quality control of raw reads?

Please specify in the method section whether what taxonomic level was used in each analysis.

In Kraken2, what confidence value did you use? Default=0.0?

MAGs completeness greater than 45% is somewhat low.

L175: how to determine abundant genera?

COG = Clusters of Orthologous Genes?

BCG=Biosynthetic gene clusters?

L159-160: what model? Linear or Linear Mixed-Effects Model? Explain the model by specifying fixed, random, interactions,..

To calculate sequencing depth (L261), the authors determine the number of reads that map to the *Drosophila* genome?

Results

Mega base pairs = Mbp or Mb

It would be good to determine whether any bacterial genera, based on Kraken2 classification, were present across the species examined. In this case, you can identify which genera can be considered part of the core microbiome.

Rarefaction curve methodology is not described in the method section. Did the authors subsample to the lowest read number in the samples?

PCoA is in the results but not in the methods. Does the R function plotMDS generate MDS or PCoA? Also, clarify the statistical analysis, such as Pearson correlation coefficient, in the methods section.

L2777-278: this information should also be mentioned in the methods section. However, the authors state that the flies were wild caught using banana traps and preserved in 100% EtOH. How did you get F1 flies?

Figure 5A: explain F, G, H, I, K,... in the caption.

Discussion

This section is relatively brief compared with the results. The authors mostly focused on certain bacteria, while there is limited or insufficient discussion on reads classification via Kraken, pathways, core microbiome, relationships between microbiome and geography, and other related topics.

Reviewer #2 (Comments for the Author):

This delightful small metagenomic study characterizes bacterial MAGs present across a sampling of *Drosophila* species. Because each fly was sequenced using nanopore and individually, the dataset provides an understanding of between fly variation as well as between species. The authors generally use standard methodologies that are well grounded, although see below for major concerns.

Major concerns:

It is surprising to not find *Acetobacter* or *Lactobacillus* here. Did you try mapping your raw reads to existing assemblies? The 10% contamination threshold is quite conservative and could have removed MAGs that were binned inappropriately or include strains of the same species (such as *Lactobacillus*).

Did you perform an agglomerate assembly across all sampled individuals? This would be the best way to ensure you have enough data to generate MAGs across all samples - then you map reads to infer abundance and also differences in structure of microbial genome content.

I would've loved to have seen more about the *Wolbachia* genomes identified here - what other *Wolbachia* are they closely affiliated with? Do they harbor the characterized Cif genes? or WOPhage?

Response to Reviewer Comments

Dear *mSystems* Eds.

Thank you for the positive and constructive reviews on our manuscript “Phylogenetic and functional diversity among *Drosophila*-associated metagenome-assembled genomes” (submitted as *mSystems*00027-25), and for the opportunity to submit a revision.

We have addressed each comment raised by the reviewers in our revised manuscript and in the responses below. In doing so, we have clarified methods, conducted additional analyses, and added a more comprehensive discussion section. Below we have listed each reviewer’s comment in *red italics* and our responses to each comment follows in black text.

We hope that you agree that our revisions have improved the quality of our manuscript to the level required by *mSystems*.

Sincerely,
Aaron Comeault, on behalf of all co-authors

Reviewer #1 (Comments for the Author):

This study by Comeault et al. about metagenomics of flies in the family Drosophilidae. They collected wild caught and F1 flies from three countries, UK, USA, Sao Tome and Principe. The study is well-designed with thorough analysis. However, the methods section, especially the statistical analyses, needs better description. The discussion could also be expanded. – Thanks for your positive appraisal of our study, and for your constructive feedback.

Introduction

L36: "microbiomes of non-model organisms"?

- Yes, that was clunky. Thanks, we’ve changed as recommended.

I think the last paragraph of the introduction is somewhat too detailed. Some information can be put in the methods and results sections.

- We have moved information about the species sampled to the methods section and condensed the ‘results’ summary of this paragraph.

Methods

The authors lacked sufficient detail in their analysis. Parameters of several programs used were not specified (e.g., default parameters,..).

- We have gone over our methods and clarified details throughout. All parameters or options used when running an analysis are all now stated where appropriate. A few relevant examples of these changes can be found starting on lines 191, 194, 202, 207, 211, and 231.

Please provide more information on the collection sites such as coordinates in Table S1. For example, North Carolina is not so informative.

- We have added the closest town or county to this table. We've avoided specific GPS locations because some flies were collected on private property (with the permission of the landowner). See updated Table S1 available at <https://doi.org/10.5281/zenodo.14173039>.

Can you please provide the key used for fly identification.

- We have added details about resources used to identify the species of flies: see lines 151-152 and lines 169 – 172.

*Why did the authors include *Z. taronus* and *Z. tsacasi* in this study because there is only one sample per species.*

- These two samples were included for comparison's sake, as they are not human commensal species. We also had previous evidence from Illumina sequencing data that these two species were infected by *Wolbachia*, so used them to see if we were able to assemble *Wolbachia* MAGs from the whole-organism reads. Because these comparisons are not central to our manuscript, we only highlight that these species are not human commensals. We have tried to clarify our sampling and sequencing on lines 139 – 172 of our revision.

L137: did the authors follow the extraction protocol exactly as described? Also the authors should cite this paper instead <https://doi.org/10.7554/eLife.66405>.

- yes, it was followed as described. The paper referenced is also cited.

How did you do quality control of raw reads?

- we did not employ qc of basecalled reads beyond the default quality score filtering employed by guppy. We've specified that all basecalled reads that passed default QC (q-score > 10) were included: see line 191-192. While not directly the same as QC on the raw reads, as it relates to qc of the set of MAGs we generated and analysed, quality control did occur in three additional places:

1) the assembler metaflye accounts for the error profile of nanopore reads during assembly, so many read-level errors will be corrected during assembly of contigs by metaflye.

2) we polished contigs assembled by metaflye with medaka. This is expected to further correct errors in assembled contigs.

3) we filtered MAGs based on CheckM completeness and quality scores. This resulted in many MAGs being removed from the final set of MAGs we analysed and helps ensure those MAGs we analysed were of high quality.

Please specify in the method section whether what taxonomic level was used in each analysis.

- In our revised version we have gone through and specified the taxonomic level being looked at for the analyses and results, when presented.

In Kraken2, what confidence value did you use? Default=0.0?

- yes, we used default parameters. We have clarified this detail to our revision (line 194).

MAGs completeness greater than 45% is somewhat low.

- we considered different cutoffs and went with 45% when we saw a similar cutoff reported in a different manuscript (which for the life of me, I can't find now). We have added some details about the variation in completeness and contamination across our focal MAGs in our revision (see Lines 264-268):

"We note that while we used cutoffs of 45% completeness and 10% contamination to identify 'focal' MAGs for subsequent analyses, 58% of these MAGs had completeness scores above 90% (82.5% with greater than 70% completeness scores) and the average contamination across these MAGs was 1.6% (see table S2 at <https://doi.org/10.5281/zenodo.14173040>)."

- We are also transparent and report CheckM completeness and contamination scores for all our MAGs in table S2 available at <https://doi.org/10.5281/zenodo.14173039>.

L175: how to determine abundant genera? – yes, we agree this phrasing was confusing. What we meant was bacterial genera for which we were able to assemble MAGs from multiple host individuals in our dataset. We have revised this section to state that we "carried out focused analyses on bacterial genera that were represented by MAGs independently assembled from multiple host individuals in our dataset (see results) using GTDB-TK's 'de_novo_wf' pipeline.." (see lines 248-249 in our revision).

COG = Clusters of Orthologous Genes? – thanks, we've clarified in our revision (line 279).

BCG=Biosynthetic gene clusters? – Thanks, we've clarified this in our revision (line 302-303).

L159-160: what model? Linear or Linear Mixed-Effects Model? Explain the model by specifying fixed, random, interactions,.. – we have clarified the RDA analysis and the terms used in our revision (see lines 210 – 225 in revision).

To calculate sequencing depth (L261), the authors determine the number of reads that map to the Drosophila genome? – Thanks, the wording was awkward here again. We did not map reads to any drosophilid genome. We've revised to clarify that are referring to the number of bacterial reads within the pool of reads derived from an individual in our data set. Lines 356 - 359 now read:

"Redundancy analysis on the taxonomic matrix revealed that the interaction between the number of bacterial sequence reads mined from the whole organism pool of sequences and the host species significantly affected bacterial diversity and abundance (model $R^2 = 81.35\%$; permutation test: $F_{9,21} = 15.54$; $P = 0.003$; see fig. S4 at <https://doi.org/10.5281/zenodo.14173040>)."

Results

Mega base pairs = Mbp or Mb

Thanks, changed to Mb.

It would be good to determine whether any bacterial genera, based on Kraken2 classification, were present across the species examined. In this case, you can identify which genera can be considered part of the core microbiome.

Thanks for this suggestion. We have run the suggested analysis and identified bacterial genera that were represented by at least 1% of reads classified as bacteria across at least half of the samples we collected (i.e. > 15 flies). This analysis identified 6 'core' genera: *Vibrio*, *Pseudomonas*, *Acinetobacter*, *Clostridium*, *Bacillus*, and *Halomonas*. We have added details of this analysis in our revision (see lines 335 - 338).

Rarefaction curve methodology is not described in the method section. Did the authors subsample to the lowest read number in the samples?

Thanks for flagging this, we've added methods for how we generated the rarefaction curves. Lines 202 - 204: "To visualize bacterial diversity within a sample we generated rarefaction curves using the `rarecurve` function from the `vegan` R library run with a step size of 500."

PCoA is in the results but not in the methods. Does the R function `plotMDS` generate MDS or PCoA? Also, clarify the statistical analysis, such as Pearson correlation coefficient, in the methods section.

We've revised our methods to clarify details of the PCoA analysis: see lines 203-207: "To summarize differences in the relative abundances of bacteria across samples we converted the `phyloseq` object to a `DGEList` using the '`phyloseq_to_edgeR`' function from the `PathoStat` library (34) and conducted a principal coordinate analysis (PCoA) using the `plotMDS` function from the `limma` R package run with the "gene.selection" option set to "pairwise".

L2777-278: this information should also be mentioned in the methods section. However, the authors state that the flies were wild caught using banana traps and preserved in 100% EtOH. How did you get F1 flies?

Indeed, the details in our methods were not comprehensive enough. We've clarified the sampling and identification procedures in the revision. See lines 148 – 167: "Individual flies were attracted to banana traps and then collected within 12 hours via aspiration or sweep netting. Flies were then briefly anesthetized with FlyNap (Carolina Biological, USA) and identified under a microscope using diagnostic traits described in "The Encyclopedia of North American Drosophilids" (29) and Yassin and David (2010) (30). Collected flies were processed in the field by placing all males and a subset of females in 100% ethanol. At some sample locations a subset of females were maintained on instant fly media to establish isofemale lines. One of the *D. immigrans* and the *Z. tsacasi* individual included in our dataset were F1 offspring of those wild females. In total, we generated sequence from 31 individual flies. Most of these individuals belonged to the 'human commensal' species *D. hydei* (N=8), *D. repleta* (N=4), *D. immigrans* (N=13), and *Zaprionus indianus* (N=4) collected from sites in the USA, UK, and São Tomé and Príncipe (see table S1 at

<https://doi.org/10.5281/zenodo.14173039>); however, for comparison, we also included one individual from each of the forest specialists *Z. taronus* and *Z. tsacasi* from São Tomé and Príncipe.”

Figure 5A: explain F, G, H, I, K,... in the caption.

We've added descriptions to the figure caption.

Discussion

This section is relatively brief compared with the results. The authors mostly focused on certain bacteria, while there is limited or insufficient discussion on reads classification via Kraken, pathways, core microbiome, relationships between microbiome and geography, and other related topics.

We have added significant discussions relating to the core microbiome, geography, and limitations to our study in the discussion. We hope you agree that this has strengthened the interpretation and placement of our results into broader context. See lines 651 – 661; 744 – 762; 791 – 798; 822– 849; 868 – 875.

Reviewer #2 (Comments for the Author):

This delightful small metagenomic study characterizes bacterial MAGs present across a sampling of Drosophila species. Because each fly was sequenced using nanopore and individually, the dataset provides an understanding of between fly variation as well as between species. The authors generally use standard methodologies that are well grounded, although see below for major concerns. – Thanks indeed for your positive comments on our manuscript!

Major concerns:

It is surprising to not find Acetobacter or Lactobacillus here. Did you try mapping your raw reads to existing assemblies? The 10% contamination threshold is quite conservative and could have removed MAGs that were binned inappropriately or include strains of the same species (such as Lactobacillus).

- We have not tried mapping reads to existing assemblies, but we did look at the proportion of sequences classified as bacteria that belonged to these two genera. This analysis found that both were rare: average of 0.18% and 0.22% of bacterial reads per sample for *Acetobacter* and *Lactobacillus*, respectively. We also looked for bacterial genera that were represented by at least 1% of reads classified as bacteria across at least half of the samples we collected (i.e. > 15 flies). This analysis identified 6 'core' genera that did not include *Acetobacter* or *Lactobacillus*: core genera: *Vibrio*, *Pseudomonas*, *Acinetobacter*, *Clostridium*, *Bacillus*, and *Halomonas*. Despite this, we were able to assemble two *Acetobacter* and 6 Lactobacillaceae MAGs. We have added a description of these MAGs: see lines 400 - 405:

“Two MAGs were classified as *Acetobacter* spp., both from the single F1 female *Z. tsacasi* sample, and 6 MAGs were classified in the family Lactobacillaceae. Five of the Lactobacillaceae MAGs were derived from flies collected from São Tomé (2 *D.*

repleta individuals, 1 *D. immigrans*, and 1 *Z. tsacasi*), and the sixth from a *Z. indianus* individual collected in North Carolina.”

The fact that the *Acetobacter* MAGs were both assembled from the *Z. tsacasi* individual in our sample, provides additional support for *Acetobacter* being rare or absent in many wild flies, as this was an F1 individual raised from a single female collected in the field. In total, our analyses provide evidence that *Acetobacter* and *Lactobacillus* may not be core members of the microbiomes of the species we study, at least with respect to abundance and prevalence across samples. This finding is supported by previous work in wild Drosophilids (see references 10, 17, and 18 in main text). We have clarified these details in our revised manuscript by adding a section to the discussion addressing this (lines 744 – 762).

Did you perform an agglomerate assembly across all sampled individuals? This would be the best way to ensure you have enough data to generate MAGs across all samples - then you map reads to infer abundance and also differences in structure of microbial genome content.

We considered this approach and decided to focus on comparisons of gene content among assembled MAGs for two reasons: 1) with 31 samples, our dataset is likely underpowered to test for functional enrichment at the microbiome level across species and sample locations. 2) We observe a significant amount of among-sample variation in the number and overall amount of bacterial sequences we were able to recover. This latter point would affect our ability to accurately predict gene abundances within drosophila-associated microbiomes. However, these constraints do not affect our ability to make comparisons between the gene content of the higher-quality MAGs we were able to generate.

We have added a discussion of limitations to our approach that highlights among-sample variation in the proportion of sequences we recovered and the limitations this might have on comprehensively characterising microbial communities to the discussion (see lines 822 - 849).

We did explore the impact of pooling reads across samples to generate MAGs for 3 of the species in our dataset; however, the ability of this approach to generate more (or higher-quality) MAGs was unclear (see table below). Again, we feel this result supports our decision to focus on comparisons among MAGs, rather than the use a standard metagenomic approach.

Species	N MAGs¹	N “good”* MAGs¹	N MAGs²	N “good”* MAGs²
D. hydei (N = 8)	51	6	19	4
D. immigrans (N = 13)	135	34	84	14
Z. indianus (N = 4)	32	4	21	5

¹total number of MAGs assembled when assembly was conducted on bacterial reads from each individual fly independently.

²MAGs assembled when assembly was conducted on the agglomerated reads across individuals of the same species.

*“good” MAGs are those with CheckM completeness > 45% and contamination < 10%

I would've loved to have seen more about the Wolbachia genomes identified here - what

other Wolbachia are they closely affiliated with? Do they harbor the characterized Cif genes? or WOPhage?

We have looked at these closer and found they are closely related to wMel-type Wolbachia. We used blast to confirm they contain both cifA and cifB. We also have added reference to a recent study that more comprehensively analysed *Wolbachia* assembled from *Z. tsacasi* and *Z. taronus*, comparing them to other *Wolbachia* isolates (Shropshire et al. 2024; <https://doi.org/10.1101/2023.12.04.569981>). We have added this information to relevant sections in the results, discussion, and SI (figure S8 below for convenience). See: lines 407-413 in the Results; lines 791-798 in the Discussion.

Fig. S8: relationships among *Wolbachia* genomes included in the GTDB-Tk database and those generated as part of our study (in bold).

Re: mSystems00027-25R1 (Phylogenetic and functional diversity among Drosophila-associated metagenome-assembled genomes)

Dear Dr. Aaron A Comeault:

Your manuscript has been accepted, and I am forwarding it to the ASM production staff for publication. Your paper will first be checked to make sure all elements meet the technical requirements. ASM staff will contact you if anything needs to be revised before copyediting and production can begin. Otherwise, you will be notified when your proofs are ready to be viewed.

Sincerely,
Mark Mandel
Editor
mSystems

Reviewer #1 (Comments for the Author):

The authors have revised the manuscript thoroughly and have addressed all the comments and suggestions. I have no further comments.

Reviewer #2 (Comments for the Author):

The authors have responded to all my prior concerns substantively. I appreciated that they went ahead and tried to use an agglomerative approach to assembly and although surprised this did not yield better MAGs, it was good to see it through.